# Dual Operation Aggregation Graph Neural Networks for Solving Flexible Job-Shop Scheduling Problem with Reinforcement Learning

## Abstract

With the widespread adoption of Internet Protocol (IP) communication technology and Web-based platforms, cloud manufacturing has become a significant hallmark of Industry 4.0. Integrating graph algorithms into these web-enabled environments is crucial as they facilitate the representation and analysis of complex relationships in manufacturing processes, enabling efficient decision-making and adaptability in dynamic environments. As a key scheduling problem in cloud manufacturing, the flexible Job-shop Scheduling Problem (FJSP) finds extensive applications in real-world scenarios. However, traditional FJSP-solving methods struggle to meet the efficiency and adaptability demands of cloud manufacturing due to generalization issues and excessive computational time, while reinforcement learning-based methods fail to learn relationships between FJSP nodes, such as interactions between operations of different jobs, leading to limited interpretability and performance. To address these issues, we propose a dual operation aggregation graph neural network (GNN) for solving FJSP. Specifically, we decouple the disjunctive graph into two distinct graphs, reducing graph density and clarifying relationships between machines and operations, thus enabling more effective aggregation and understanding by neural networks. We develop two distinct graph aggregation methods to minimize the influence of non-critical machine and operation nodes on decision-making while enhancing the model's ability to account for long-term benefits. Additionally, to achieve more accurate multi-objective estimation and mitigate reward sparsity, we design a reward function that simultaneously considers machine efficiency, schedule balance, and makespan minimization. Extensive experimental results on well-known datasets demonstrate that our model outperforms state-of-the-art models and exhibits excellent generalization capabilities, effectively addressing the challenges of cloud manufacturing.

## CCS Concepts

• **Networks** → **Network algorithms**; • **Computing methodologies** → **Machine learning**; **Artificial intelligence**.

## Keywords

Flexible Job-Shop Scheduling Problem, Graph Neural Network, Reinforcement Learning, Combinatorial Optimization

## 1 Introduction

The flexible job-shop scheduling problem (FJSP) is a well-known NP-hard problem [50], extending the job-shop scheduling problem with a significantly larger solution space and increased complexity [1, 10]. The FJSP is characterized by operations with multiple processing options, constrained by processing sequences and machine load [7].

The objective of solving FJSPs is to minimize the makespan [41], which is typically pursued in the manufacturing industry [24].

With the widespread adoption of Internet Protocol (IP) communication technology, web-based cloud manufacturing provides an intelligent and efficient development trajectory for manufacturing systems [3], necessitating algorithms with adaptive and efficient problem-solving capabilities [40]. In this context, the integration of smart online scheduling becomes crucial. The FJSP presents significant opportunities for time and cost savings in industries such as aerospace engine manufacturing [57] and semiconductor production [12] when effectively solved within cloud manufacturing environments [14]. By leveraging network structures, we can better exploit the complex relationships and interactions inherent in FJSPs. Utilizing graph-based models allows for the representation of operations and machines as nodes, while their interactions as edges, facilitating insightful analysis and enhanced solution strategies.

Due to the importance of the FJSP in both practical application and theoretical study, numerous traditional methods have been proposed to solve FJSPs, which are broadly categorized into three types: exact methods, dispatching rules and metaheuristic methods. However, these methods often result in insufficient scheduling quality [37, 49], limited generalization capabilities [15, 42], and time-consuming processes [54]. Therefore, traditional methods cannot fully leverage the advantages of graph-based representations, making it difficult to meet the demands of cloud manufacturing.

As a promising alternative, the methods based on reinforcement learning (RL) have been widely employed in recent years to solve FJSPs [11, 15, 51]. These methods often utilize Markov Decision Processes (MDPs) [23, 26], where the agent assesses decision quality by computing the reward garnered from each decision and then improves the agent's policy iterative through maximizing rewards. Furthermore, these RL-based methods are usually integrated with graph neural networks (GNNs) designed based on disjunctive graphs [8, 37, 54], since these graphs indicate the relationships among operations and machines in the scheduling process [2].

However, current methods suffer from two significant limitations. Firstly, disjunctive graph-based GNNs often fall short in capturing comprehensive relationships. Specifically, disjunctive graphs cannot capture the logical relationships between operations from different jobs, because the directed arcs do not connect these operations, but agents must select operations belonging to different jobs during each decision-making process. The policy's failure to consider these logical relationships adversely affects the agent's decision quality. Secondly, the reward functions of the existing methods face challenges in accurately reflecting the agent's decision quality. Typically, the reward is defined as the difference between the minimum makespan before and after making a decision [37, 51, 53]. However, this approach often involves multiple

reward estimates when computing the makespan during the scheduling process, leading to estimation error accumulation, which hinders accurate guidance for the scheduling process. Furthermore, since this approach is designed merely based on the final scheduling objective, it is challenging to reflect specific decisions before the terminal state, resulting in a sparse reward [38].

To address the aforementioned limitations, we present substantial advancements in the model and the reward function. Firstly, we introduce a model named Dual Operation Aggregation Graph Neural Networks (DOAGNN) aimed at capturing the relationships between machines and operations. Specifically, we decompose the disjunctive graph into two distinct graphs to effectively represent the relationships between machines and operations, as well as the interactions among different operations. Subsequently, we design two tailored GNNs for these graphs to integrate global scheduling information into actionable operations, mitigating the impact of machines with long processing times and late-scheduled operations. Additionally, we develop new aggregation methods to facilitate information propagation between operations of different jobs, enhancing the agent's ability to consider long-term benefits. Secondly, we redesign the reward function in the MDP. Particularly, besides minimizing makespan as used in previous studies [37, 51, 53], we add two components to the reward to increase the minimum completion time and reduce machine idle time, aimed to mitigate the impact of estimated makespan on decision evaluation, enhance the feedback of decisions on scheduling objectives, and alleviate reward sparsity. In summary, our contributions are stated as follows:

- We propose a method to decouple the complex disjunctive graph into two distinct graphs, accurately representing the scheduling state and dependencies among operations across different jobs, thus aiding subsequent neural networks in better understanding and solving FJSPs.
- We propose an innovative model named DOAGNN for solving FJSPs, featuring two tailored GNNs to more comprehensively and effectively aggregate information from both machines and operations.
- We develop a reward method that delivers real-time feedback on the impact of decisions on scheduling objectives, reducing biases from estimation uncertainties and alleviating reward sparsity.
- We demonstrate excellent performance of DOAGNN by conducting extensive experiments with seven state-of-the-art baselines on well-known datasets.

## 2 Related works

In this section, we introduce traditional methods and RL-based methods for solving FJSPs.

### 2.1 Traditional Methods

Traditional methods for solving FJSPs can be broadly categorized into three types: exact methods, dispatching rules and metaheuristic methods. Exact methods systematically explore the entire solution space to achieve the optimal solution, exemplified by the branch and bound algorithm [5]. Dispatching rules select machines and operations for processing based on pre-designed scheduling rules [16], such as selecting the machine with the shortest processing time.

Metaheuristic methods rely on carefully designed algorithms to explore the solution space, often yielding high-quality solutions. Consequently, numerous metaheuristic methods have been applied to better solve FJSPs, such as the genetic algorithm [32] and the particle swarm optimization algorithm [30, 43, 46]. Additionally, researchers also utilize RL to adjust the parameters of metaheuristic methods, aiming to improve solution quality [9, 22, 25, 44].

However, traditional FJSP solvers struggle to meet the requirements of cloud manufacturing. Specifically, exact methods face challenges in efficiently exploring the solution space of large-scale FJSPs within a reasonable time [48, 54]. Dispatching rules that rely on simplistic scheduling rules struggle to capture complex scheduling principles, thereby impeding the generation of high-quality solutions [37]. Metaheuristic methods are time-consuming and fall short in generalization capabilities [6, 15, 42].

### 2.2 Reinforcement Learning-Based Methods

In recent years, numerous RL-based methods have been proposed to solve FJSPs, with predominant approaches involving the integration of RL with GNNs [4, 15, 27, 37, 47, 54]. RL is responsible for making decisions of scheduling, while GNNs focus on embedding and aggregating features based on the disjunctive graph.

The RL-based methods for solving FJSPs utilize a hierarchical structure to select the operations and machines separately. Brandimarte et al. [4] initially introduced this hierarchical concept, implementing it with tabu search. The subsequent methods typically utilize RL to determine selections, and treat the selections as decisions in each step of the MDP [20, 21, 28, 29, 45]. However, the information about operations and machines is interactive, leading agents to overlook the interplay between selecting operations and machines during decision-making.

In addition to exploring various scheduling strategies, researchers have employed various RL-based algorithms to solve FJSPs, such as the Proximal Policy Optimization (PPO) algorithm [55, 58] and the Deep Deterministic Policy Gradient algorithm [13]. Furthermore, researchers have endeavored to employ multi-agent reinforcement learning methods to solve FJSPs [19, 31]. However, RL-based algorithms do not ensure that agents learn the accurate logical relationships in the scheduling process because the FJSP is highly complex and nonlinear. Addressing this challenge necessitates acquiring appropriate embeddings and aggregating features of machines and operations from GNNs.

As GNNs are designed based on the disjunctive graph, researchers have developed various graphs to represent the scheduling process of the FJSP, thereby determining the direction of feature aggregation. Han et al. [15] proposed a three-dimensional disjunctive graph, while Song et al. [37] transformed disjunctive graph into a heterogeneous graph. Subsequently, Zhang et al. [54] designed a multi-agent graph for the FJSP, capturing the relationships between machines and operations. However, these approaches rely heavily on the sequential scheduling logic of the FJSP, leading to overly complex and dense graph structures that hinder model learning. Moreover, they overlook relationships among operations from different jobs, causing each operation to ignore meaningful changes in the scheduling environment.

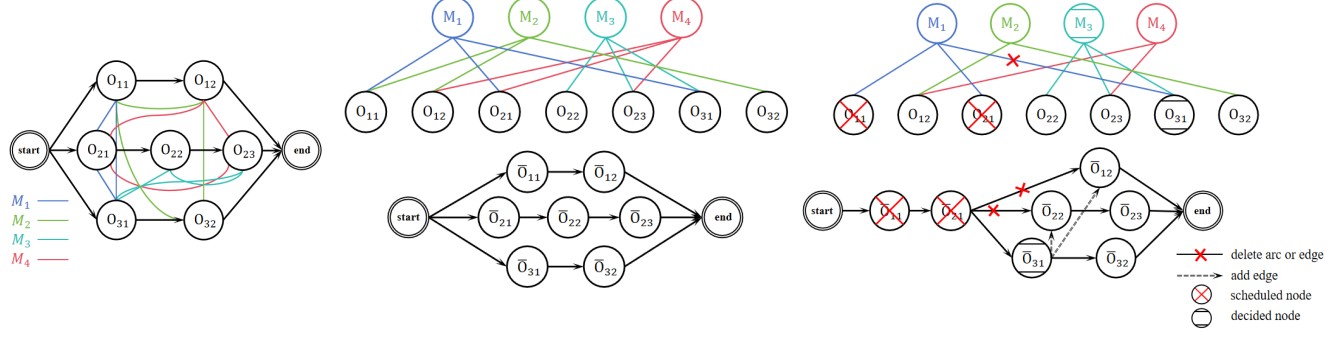

(a) Disjunctive graph.   (b) M2O graph (up) and O2O graph (down).   (c) Arc changes of M2O graph (up) and O2O graph (down).

**Figure 1: The illustrations of disjunctive graph, the initial M2O graph and O2O graph, and their arc changes.**

## 3 Preliminaries

In this section, we define the fundamental setting and the disjunctive graph of the FJSP.

### 3.1 The FJSP Setting

The FJSP comprises a series of jobs and machines, with each job possesses its own sequence of operations. Each operation must be processed on a specified set of machines. The scheduling process is complete when all operations have been assigned to machines.

We define a set of $n$ jobs as $J = \{J_1, J_2, ..., J_n\}$, where $J_i$ denotes the $i$th job. Similarly, we define a set of $m$ machines as $M = \{M_1, M_2, ..., M_m\}$, where $M_k$ denotes the $k$-th machine. The job $J_i$ consists of $n_i$ consecutive operations, which are denoted as the set $O_i = \{O_{i1}, O_{i2}, ..., O_{in_i}\}$. $M_{ij}$ denotes the set of machines capable of processing operation $O_{ij}$, and the processing time of operation $O_{ij}$ on machine $M_k$ ($M_k \subseteq M_{ij}$) is denoted as $p_{ijk}$. We denote the completion time of operation $O_{ij}$ as $C_{ij}$, and $C_{in_i}$ denotes the completion time of the last operation $O_{in_i}$ of job $J_i$, which corresponds to the makespan of job $J_i$. Our optimization goal is to minimize the makespan $C_{max} = \max(C_{in_i}), \forall i \in \{1, 2, ..., n\}$.

The FJSP satisfies the following constraints during scheduling: 1) operations within the same job must be processed sequentially; 2) operations must be processed within the specified machine sets; 3) once an operation starts processing, it cannot be interrupted; 4) each machine processes only one operation at a time.

### 3.2 Definition of The Disjunctive Graph

We denote the scheduling state of the FJSP as a disjunctive graph $G = \{O, C, D\}$. Here, $O = \{O_i | \forall i \in \{1, 2, ..., n\}\} \bigcup \{S, E\}$, where $S$ and $E$ denote the start and end virtual nodes, respectively. $C$ denotes the directed arcs connecting adjacent operations on the same jobs, and $E$ denotes the undirected disjunctive edges connecting adjacent operations on the same and compatible machines. During the scheduling process, the direction for each disjunctive arc within an operation node needs to be determined, and the remaining undirected disjunctive edges are removed. We present an illustration of a disjunctive graph in Figure 1a.

## 4 Method

In this section, we introduce DOAGNN, an advanced model designed to effectively solve FJSPs. DOAGNN features two GNNs tailored for the decoupled disjunctive graph, embedding machine features and other operation features into operation nodes separately. At each step of the MDP, the agent utilizes the output from DOAGNN to make decisions (i.e., select the appropriate machine and operation) until the scheduling process is complete. The following subsections provide a detailed explanation of the MDP for the FJSP, the decomposition and improvement of the disjunctive graphs, the architecture of the DOAGNN, and the training process.

### 4.1 Markov Decision Process

We define the process of solving FJSPs as a discrete MDP. At each step, the agent makes a decision by selecting from the action space, and the state transitions based on this decision. The scheduling process is completed when all operations are scheduled. In the following paragraphs, we elaborate on the details of the MDP, including the novel settings of state, action and reward.

*4.1.1 State.* We define the state as the current scheduling situation in the FJSP. Specifically, at each step $p$ of the MDP, the state $s_p$ comprises the states of both operations and machines (refer to Appendix A for details of the composition of states). When an operation is not scheduled, the start time $T_s$ and end time $T_e$ in this operation state are estimated. We define the estimated time $T_s$ and $T_e$ of the unscheduled operation $O_{ij}$ at step $p$ as follows:

$$T_s(O_{ij}, s_p) = \max(T_e(O_{i,j-1}, s_p), \sum_{M_k \in M_{ij}} T_m(M_k, s_p)/|M_{ij}|), \quad (1)$$

$$T_e(O_{ij}, s_p) = T_s(O_{ij}, s_p) + \sum_{M_k \in M_{ij}} p_{ijk}/|M_{ij}|, \quad (2)$$

where $T_m(M_k, s_p)$ denotes the actual processing end time of machine $M_k$ in the state $s_p$. For the initial state, we set $T_s$ and $T_m$ to 0. The start and end time of scheduled operations adhere to their actual time, while the start and end time of unscheduled operations are iteratively computed according to (1) and (2).

*4.1.2 Action.* We define an action as the decision that the agent makes at a certain step. Considering that separating the decision-making process for operations and machines overlooks the impact of the machine state on the decisions of operations, we simultaneously determine both the operation and machine at each step. Consequently, the action space includes all combinations of schedulable operations and available machines.

*4.1.3 Reward.* We consider three objectives when setting the reward: minimizing the makespan, maximizing the minimum completion time, and minimizing machine idle time. Firstly, the most common scheduling objective of the FJSP is to minimize the makespan (i.e., $\min(C_{max})$), we define the difference between the $C_{max}$ before and after the state transition as the first objective of our reward as:

$$r_{tgt}(s_p, a_p, s_{p+1}) = \widehat{C}_{max}(s_p) - \widehat{C}_{max}(s_{p+1}), \tag{3}$$

where $\widehat{C}_{max}(s_p)$ denotes the estimated value of $C_{max}$ in the state $s_p$, computed by the minimum estimated end time $T_e$ of the last operations of all jobs. Considering this reward requires reestimation at each time step, we introduce two additional directly measurable objectives to balance and mitigate the impact of estimation, encouraging more comprehensive and practical decisions.

In addition to the scheduling objective of $\min(C_{max})$, we consider simultaneously increasing both maximum makespan and minimum completion time, and reducing machine idle time (the details of analysis are described in Appendix B). When considering these two objectives, machines are evenly selected for processing, while maximizing the load on each machine as much as possible. We also define these two objectives $r_{ave}$ and $r_{util}$ as reward:

$$r_{ave}(s_p, a_p, s_{p+1}) = \min(C_{ij}(s_p)) - \min(C_{i'j'}(s_{p+1})), \tag{4}$$

$$r_{util}(s_p, a_p, s_{p+1}) = \max(\mu_{ijk'} C_{ij}(s_p)) - C_{i'j'}(s_{p+1}), \tag{5}$$

where $ij$ denotes the index of any operation in the set of all scheduled operations. $\mu_{ijk}$ denotes a decision variable with the value equal to 1 when operation $O_{ij}$ is processed by machine $M_k$, and 0 otherwise. $O_{i'j'}$ and $M_{k'}$ represent the operation and machine selected by the agent at the current step, respectively. According to (4) and (5), the agent receives higher rewards as the minimum completion time increases, while rewards decrease with longer machine idle time. Finally, we define the total reward as:

$$r = \lambda_1 r_{tgt} + \lambda_2 r_{ave} + \lambda_3 r_{util}, \tag{6}$$

where the hyperparameters $\lambda_i \in (0, 1]$ ($\forall i \in \{1, 2, 3\}$) balance the numerical contributions of the three reward components (the analysis of hyperparameters settings is provided in the Appendix C). Furthermore, we calculate the discounted return $G_p$ at step $p$ as follows:

$$G_p = \sum_{k=0}^{-p+\sum_{i=1}^{n} n_i} \gamma^k r_{p+k}, \tag{7}$$

where $r_i$ denotes the actual reward obtained at step $i$ of the MDP, and the discount factor $\gamma \in (0, 1)$ denotes the degree of emphasis on long-term returns.

## 4.2 The Decomposition and Improvement of Disjunctive Graphs

To reduce graph density and clarify the relationships between machines and operations, we decompose the disjunctive graph into two separate graphs. As shown in Figure 1b, one graph illustrates the

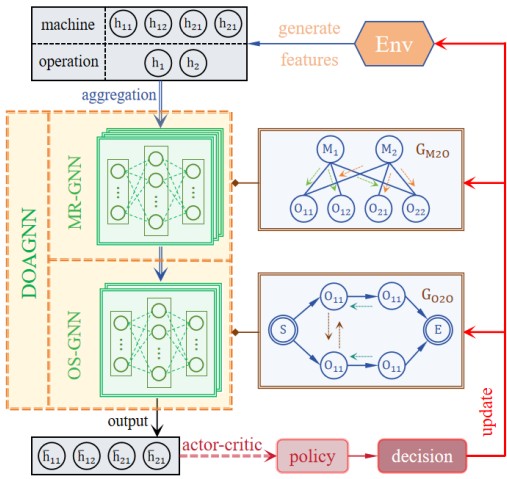

**Figure 2: The framework of DOAGNN.**

relationships among machines and operations, named the "Machine to Operation graph" (M2O graph). Another graph illustrates the relationships among the different operations, named the "Operation to Operation graph" (O2O graph). In the M2O graph, operations and machines are represented through nodes, and these nodes are connected by undirected edges, which store the corresponding processing time information. These undirected edges indicate the compatibility between operations and machines, rather than describing the processing sequence relationships for the same machine among different operations, as traditional disjunctive arcs do [36]. The O2O graph retains the backbone of directed arcs from the disjunctive graph, and effectively indicates the processing sequences among operations through a specially designed edge update method (see Section 4.3.2 for details).

Formally, we define the M2O graph and O2O graph as $G_{M2O} = (O, M, P)$ and $G_{O2O} = (\bar{O}, C)$, respectively. Here, $\bar{O}$ denotes the set of nodes containing machine information corresponding to the operation set $O$, and $P$ denotes the undirected edge set connecting machines and operations. Both $P$ and $C$ vary with changes in the state. Specifically, when making decision ($O_{ij}$ and $M_k$) in state $s_p$, the edges in $P$ connected with the nodes ($O_{ij}$ and $M_{k'}$ ($\forall k' \in \{1, ..., m\}, k' \neq k$)) are removed, indicating that the machine selection for the operation $O_{ij}$ is determined. Then, new directed arcs are added in $C$, pointing from $\bar{O}_{ij}$ to $\bar{O}_{i'j'}$ ($\forall \bar{O}_{i'j'} \in \bar{O}_p$, $\bar{O}_p$ denotes the set of all nodes with the same predecessor as $\bar{O}_{ij}$), while simultaneously removing all directed arcs from the predecessor of $O_{ij}$ to $O_{i'j'}$. This signifies that the decision operation node is connected to all possible decision operation nodes in the next state. We present an illustration of arc changes of the M2O graph and O2O graph in Figure 1c.

## 4.3 DOAGNN Model

We propose a model named DOAGNN to enrich the embedding information of operation features, utilizing two tailored GNNs based on the M2O graph and O2O graph. As shown in Figure 2, the architecture of DOAGNN includes: Machine Relation GNN (MR-GNN) to

learn the embeddings between different machines and their corresponding operations using the M2O graph, and Operation Sequence GNN (OS-GNN) to learn the embeddings between operation features that include machine information using the O2O graph.

### 4.3.1 MR-GNN.

To enhance the connection between machines and operations, we propose MR-GNN, which embeds machine features into operation features. In $G_{M2O}$, the neighborhood set $N_{ij}$ of any operation node $O_{ij}$ comprises exclusively available machine nodes. We denote $h_{ij}$ and $h_k$ as the features of operation $O_{ij}$ and machine $M_k$, respectively. The initial features are encoded from the operation states and machine states. We embed the features of all machines in the neighborhood set $N_{ij}$ into the feature of the corresponding operation node $O_{ij}$. Drawing inspiration from the Relational Graph Convolutional Network (RGCN) that handles various relationships among graph nodes [33], we consider the type of each machine as the embedding relation for the corresponding operation. The feature $\bar{h}_{ij}$ of the operation node $O_{ij}$ after embedding is defined as:

$$\bar{h}_{ij} = \text{LeakyReLU}(\sum_{r \in R_m} \sum_{k \in N_{ij}^r} d_{ijk} W_r h_k + W_0 h_{ij}), \tag{8}$$

where $R_m$ represents the set of relation types, corresponding to the number of machines $m$. The learnable parameters $W_0$ and $W_r$ ($r \in R_m$) capture these relationships. However, as the number of machines increases, the parameter matrix $W_r$ grows proportionally, affecting model training efficiency. To address this, we apply basis decomposition to reduce the dimensionality of $W_r$ as follows:

$$W_r = \sum_{b=1}^{B} a_{rb} V_b, \tag{9}$$

where the hyperparameter $B$ denotes the number of decomposed blocks, $V_b$ denotes the parameter matrix shared among all relations, and $a_{rb}$ denotes the parameter matrix specific to each relations.

RGCN does not consider the information of undirected edges. To address this, we convert the processing time $p_{ijk}$ associated with undirected edges into a distance $d_{ijk} \in (0, 1)$ between the connected operation and machine nodes. Longer processing time causes the distance between operation and machine nodes to approach 0. Consequently, machines with longer processing time exert less influence on the operation nodes during embedding. We define the computation of the distance $d_{ijk}$ as follows:

$$d_{ijk} = (||W_d(h_k/p_{ijk}), h_{ij}||_{\cos} + 1)/2|N_{ij}|, \tag{10}$$

where the parameter $W_d$ learns the similarity between machine nodes and operation nodes. $||a, b||_{\cos}$ denotes the computation of the cosine similarity between vectors $a$ and $b$. The value of processing time $p_{ijk}$ serves as the scaling factor for the machine feature, which mitigates the impact of excessively long processing time on the features of the corresponding operation. $|N_{ij}|$ denotes the number of elements within the neighborhood set $N_{ij}$.

### 4.3.2 OS-GNN.

Through MR-GNN, operation nodes are effectively embedded the features of their neighboring machine nodes. Since the agent makes decisions by selecting from all schedulable operation nodes, and due to the sequential scheduling constraints of the FJSP, these operations necessarily belong to different jobs. Therefore, we aggregate global information into the schedulable operation nodes through $G_{O2O}$.

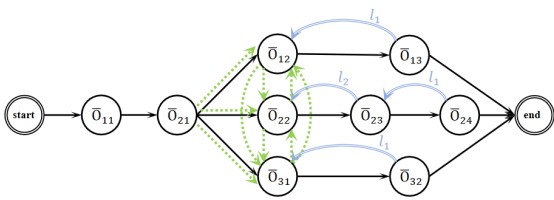

**Figure 3: An illustration of the directions of node information aggregation. The green lines denote the direction of the same job's operations, and the blue lines denote the direction of the different job's operations.**

The classical disjunctive graph does not include directed arcs connecting operation nodes of different jobs, resulting in the embeddings based on the arcs failing to capture the impact of these operations on the current operation. Therefore, we propose two new aggregation methods to address this issue. Specifically, operation nodes from the same job transmit information in the direction opposite to the directed arcs. We denote the last two nodes scheduled from the same job as the first layer, with layers incrementing forward in the reverse direction of the directed arcs until the next qualified scheduling node. After aggregating information from operations of the same job, we aggregate information from operations of different jobs. To prevent redundant aggregation, the information of aggregated node do not transmit further.

We present an instance in Figure 3 to illustrate the direction of information aggregation. As shown in Figure 3, the schedulable operations are $O_{12}, O_{22}, O_{31}$. Firstly, these operations aggregate information from successor nodes that are hierarchically layered (i.e., $\bar{O}_{13}, \bar{O}_{23}, \bar{O}_{24}, \bar{O}_{32}$). Secondly, the nodes aggregate information from the operations belonging to different jobs and the previously scheduled. When aggregating information for $\bar{O}_{12}$, $\bar{O}_{12}$ also incorporates the information for $\bar{O}_{22}, \bar{O}_{31}$ and $\bar{O}_{21}$. The aggregation for $\bar{O}_{22}$ and $\bar{O}_{31}$ follows the same process as for $\bar{O}_{12}$.

According to the directions of two different aggregations, we design two aggregation computation methods for operation nodes from the same job and from different jobs, respectively.

For two neighboring nodes $\bar{h}_{i,j}$ and $\bar{h}_{i,j+1}$ from the same job at the $l$-th layer, we define the aggregation computation as follows:

$$\bar{h}_{i,j}^{(l+1)} = W_n(\bar{h}_{i,j}^{(l)} + \gamma_o \bar{h}_{i,j+1}^{(l)})/2, \tag{11}$$

where the parameter $W_n$ learns the feature aggregation for neighboring nodes, and the discount factor $\gamma_o$ aims to diminish the influence of operations scheduled later on the current operation node. We define the computation of feature $\bar{h}_{i,j}$ for the operation node awaiting scheduling in the $l$th layer as follows:

$$\bar{h}_{i,j}^l = \sigma((\gamma_o W_n/2)^{l-1} \bar{h}_{i,m+l-1}^{(1)} + \sum_{k=1}^{l-1} \gamma_o^{k-1} (W_n/2)^k \bar{h}_{i,m+k-1}^{(l-k)}). \tag{12}$$

For the operation node feature $\bar{h}_{i,j}$ and the feature set $\bar{h}' = \{\bar{h}_1, \bar{h}_2, ..., \bar{h}_{n'}\}$ composed of the other schedulable operation nodes, we define the aggregation computation as follows:

$$\bar{h}_{i,j} = \text{LeakyReLU}(W_d \cdot \text{mean}(\bar{h}_{i,j}||\gamma_o \bar{h}_1||...||\gamma_o \bar{h}_{n'})), \tag{13}$$

**Algorithm 1:** Aggregate the operation features in $G_{O2O}$

**Input:** $G_{O2O} = (\bar{O}, C)$, set of indices for $n$ schedulable
     nodes $\bar{O}_L$, discount factor $\gamma_o$
**Output:** schedulable node feature set $L$

1   $L_n \leftarrow [\,]$ // save the embedded features
2   **for** $i \leftarrow 1$ **to** $n$ **do**
3      $\bar{O}_i \leftarrow$ features of the $i$th node and its successor nodes
4      **for** $i_o \leftarrow \text{len}(\bar{O}_i)$ **to** $2$ **do**
5         $L_n.\text{append}(\textbf{Aggeration\_1}(\bar{O}_i[i_o], \bar{O}_i[i_o - 1], \gamma_o))$
           // according to Eq.(11)
6      **end**
7   **end**
8   **for** $iter_1 \leftarrow 1, iter_2 \leftarrow 1$ **to** $n$ **do**
9      $L.\text{append}(\textbf{Aggeration\_2}(L_n[iter_1], L_n[iter_2]))$
         // according to Eq.(13)
10 **end**

where $a||b$ denotes the concatenation of vectors $a$ and $b$. The parameter $W_d$ learns the embedding of the operation features from different jobs, and the discount factor $\gamma_o$ is designed to mitigate the influence of other nodes to be aggregated on the current node.

In summary, we mitigate the impact of operations scheduled late by aggregating information from operations within the same job into schedulable operations. The aggregation across different jobs treats each schedulable operation as the next to be scheduled, incorporating long-term benefits into the aggregated information. The embedding algorithm is detailed in Algorithm 1.

### 4.4 Training Process

We adopt an actor-critic framework [39] to train our model, where the actor guides the agent to select machines and operations to schedule via the policy network, while the critic evaluates the value of the agent's decision via the value network.

The actor generates decision policy and selects actions. First, we concatenate the output operation features from OS-GNN to form the feature $h_{skd}$ as $[\bar{h}_1||\bar{h}_2||...||\bar{h}_{n'}]$. Next, we pass $h_{skd}$ into policy network to determine the action $a_p$ in the current state $s_p$. The computation for the policy network $\pi(a|s;\theta)$ is defined as:

$$\pi(a_p|s_p;\theta) = \frac{e^{S \odot (W_\theta \cdot \text{fl}(h_{skd}))}}{\sum_1^n \sum_1^m e^{S \odot (W_\theta \cdot \text{fl}(h_{skd}))}}, \quad (14)$$

where $\text{fl}(a)$ flattens the feature matrix $a$ along the first dimension. $S$ is a set of masks determining machine availability for scheduling. $S_i$ represents a row-wise slice of $S$, with elements set to 1 if the corresponding machine is idle, and 0 otherwise. $\odot$ denotes element-wise multiplication. The learnable parameter $W_\theta$ consists of a Multilayer Perceptron (MLP) with a single hidden layer and two non-linear activation layers.

The critic evaluates the value $V$ of the policy by the value network. We define the value network $V(s;\omega)$ as follows:

$$V(s_p;\omega) = W_\omega(\text{pool}(h_{skd})), \quad (15)$$

where the function pool denotes the averaging pooling method applied to each operation feature. The parameters $W_\omega$ are constructed similarly to $W_\theta$ except for the last layer.

The objective of the actor is to maximize the mathematical expectation of the return obtained by policy. We utilize Generalized Advantage Estimation (GAE) [34] to estimate the advantage function $A_p$ for state $s_p$ as the total return, as follows:

$$A_p = \sum_{k=0}^{\infty} (\gamma\lambda)^k (r_{p+k+1} + \gamma V(s_{p+k+1};\omega) - V(s_{p+k};\omega)), \quad (16)$$

where $\lambda$ denotes the decay parameter to reduce variance and bias. We employ PPO algorithm [35] to limit the step size of policy updates. We define the loss function $L_\theta$ of policy network as follows:

$$L_\theta = \frac{1}{N} \sum_{i=1}^{N} \min(r_{p_i}(\theta)A_{p_i}, \text{Clip}(r_{p_i(\theta)}, 1 - \epsilon, 1 + \epsilon)A_{p_i}), \quad (17)$$

$$r_{p_i}(\theta) = \exp(\log \pi(a_p|s_p;\theta) - \log \pi(a_p|s_p;\theta_{old})), \quad (18)$$

where $N$ denotes the total number of samples, $p_i$ denotes the current step of the $i$th sample, $r_{p_i}(\theta)$ denotes the probability ratio of the $i$th sample before and after policy update. $r_{p_i}(\theta)$ calculates the probability ratio by the log probability as shown in Eq. (18). The function Clip constrains the probability ratio to the $[1 - \epsilon, 1 + \epsilon]$ interval, where $\epsilon$ is a hyperparameter.

The objective of the critic is to minimize the difference between the estimate of the value network and the actual return. We define the loss function $L_\omega$ of value network as follows:

$$L_\omega = \frac{1}{N} \sum_{i=1}^{N} (V(s_{p_i};\omega) - G_{pi})^2. \quad (19)$$

## 5 Experiments

This section presents a comprehensive evaluation of our method. We begin by detailing the experimental configuration, baseline models, and benchmark datasets used in our experiments. Then, we compare the performance of DOAGNN to the baseline models. Next, we assess the generalization capability of DOAGNN across FJSPs of varying sizes. Finally, we conduct ablation studies focusing on our DOAGNN and reward function.

### 5.1 Basic Problem Statement

In this subsection, we sequentially introduce the experimental configuration in our model training, describe the selected baselines, and detail the composition of the selected benchmarks.

*5.1.1 Experimental Configuration.* All experiments are conducted on a platform consists of an Intel I9-12900K processor with 24 cores, 64GB memory, and an RTX4090 GPU with 24GB VRAM. We present the detailed parameter settings in Appendix C.

*5.1.2 Baselines.* We employ seven state-of-the-art (SOTA) RL-based methods, two heuristic methods and four dispatching rules for solving FJSPs as baselines for performance comparisons.

The following is a brief overview of the seven selected RL-based methods. Han et al.[15] proposed a Modified Pointer Network (MPN) for solving FJSPs. Feng et al.[11] introduced an actor-critic framework optimized with PPO to tackle FJSPs, termed PPO4FJSP. Zeng et al.[52] developed a Graph Isomorphism Network (GIN) to extract operation features, optimizing the model using the Asynchronous Advantage Actor-Critic algorithm (MTA3C). Lei et al.[21] presented a Multi-Pointer Graph Network (MPGN) for solving FJSPs.

**Table 1: Performance of our DOAGNN on the Brandimarte dataset.**

| $n \times m$ | $10 \times 6$ | $10 \times 6$ | $15 \times 8$ | $15 \times 8$ | $15 \times 4$ | $10 \times 15$ | $20 \times 5$ | $20 \times 10$ | $20 \times 10$ | $20 \times 15$ | |
|---|---|---|---|---|---|---|---|---|---|---|---|
| | MK01 | MK02 | MK03 | MK04 | MK05 | MK06 | MK07 | MK08 | MK09 | MK10 | AVE GAP |
| LB | *40* | *26* | *204* | *60* | *172* | *58* | *139* | *523* | *307* | *198* | |
| MWKR×EET | 51(27.5%) | 41(57.7%) | 210(2.9%) | 99(65%) | 202(17.4%) | 112(93.1%) | 215(54.7%) | 579(10.7%) | 384(25.1%) | 291(47%) | 26.5% |
| MWKR×LWT | 57(42.5%) | 41(57.7%) | 234(14.7%) | 90(50%) | 211(22.7%) | 114(96.6%) | 219(57.6%) | 631(20.7%) | 397(29.3%) | 294(48.5%) | 32.5% |
| MOPNR×EET | 49(22.5%) | 42(61.5%) | 219(7.4%) | 84(40%) | 201(16.9%) | 107(84.5%) | 220(58.3%) | 537(2.7%) | 355(15.6%) | 286(44.4%) | 21.6% |
| MOPNR×LWT | 51(27.5%) | 43(65.4%) | 230(12.8%) | 83(38.3%) | 192(11.6%) | 118(103.5%) | 227(63.3%) | 551(5.4%) | 381(24.1%) | 302(52.5%) | 26.1% |
| MPN [15] | 44(10%) | 28(7.7%) | 245(20.1%) | 74(23.3%) | 193(12.2%) | 123(112.1%) | 216(55.4%) | **523(0%)** | 386(25.7%) | 337(70.2%) | 25.6% |
| PPO4FJSP [11] | 42(5%) | 32(23.1%) | **204(0%)** | 78(30%) | 187(8.7%) | 90(55.1%) | 169(21.6%) | 531(1.5%) | 349(13.7%) | 279(40.9%) | 13.6% |
| MTA3C [52] | 48(20%) | 34(30.8%) | 235(15.2%) | 77(28.3%) | 192(11.6%) | 78(34.5%) | 190(36.7%) | 544(4%) | 375(22.2%) | 256(29.3%) | 17.49% |
| MPGN [21] | 47(17.5%) | 30(15.4%) | **204(0%)** | 76(26.7%) | 178(3.5%) | 79(36.2%) | **152(9.4%)** | 541(3.4%) | 335(9.1%) | **236(19.2%)** | 8.7% |
| HGNN [37] | 44(10%) | 31(19.2%) | 211(3.4%) | 78(30%) | 183(6.4%) | **74(27.6%)** | 156(12.2%) | 524(0.2%) | 326(6.2%) | 241(21.7%) | 8.2% |
| DRL-AC [56] | **41(2.5%)** | **28(7.7%)** | 206(1%) | 88(46.7%) | **175(1.74%)** | 93(60.3%) | 213(53.2%) | 525(0.4%) | 361(17.6%) | 277(40%) | 16.2% |
| LMDRL [51] | 49(22.5%) | 43(65.4%) | 216(5.9%) | 75(25%) | 190(10.5%) | 103(77.6%) | 212(52.5%) | **523(0%)** | 349(13.7%) | 264(33.3%) | 17.2% |
| **DOAGNN** | **41(2.5%)** | 33(26.9%) | **204(0%)** | **66(10%)** | 177(2.9%) | 83(43.1%) | 174(25.2%) | **523(0%)** | **311(1.3%)** | 244(23.2%) | **7.4%** |

Song et al.[37] designed a Heterogeneous Graph Neural Network (HGNN) to manage embeddings of machine and operation features. Zhao et al.[56] proposed a Deep Reinforcement Learning (DRL) framework based on an actor-critic architecture (DRL-AC) for FJSPs. Yuan et al. [51] utilized a lightweight MLP within a DRL framework (LMDRL) and employed PPO to optimize model parameters.

The two selected heuristic methods are recent genetic algorithm and jaya algorithm. Caldeira et al. [6] integrate jaya algorithm with a local search technique and an acceptance criterion (IJA) to solve FJSPs. Rooyani et al. [32] develop an efficient two-stage genetic algorithm (2SGA) for FJSPs.

The dispatching rules used include four types: MWKR × EET, MOPNR × EET, MWKR × LWT, and MOPNR × LWT. MWKR and MOPNR represent selecting jobs with the most work remaining and the most operations remaining, respectively. EET and LWT represent selecting machines with the earliest end time and the least total processing time, respectively.

*5.1.3 Benchmarks.* We utilize two publicly available datasets: Hurink dataset [18], and Brandimarte dataset, serving as benchmarks in accordance with previous studies [11, 15, 21, 37, 51, 52, 56]. Hurink dataset comprises three sets of data with different scales (Edata, Rdata, Vdata), where the scale refers to the average number $\overline{num}$ of operations executable on each machine with $\overline{num}_{\text{Edata}} = 1.15$, $\overline{num}_{\text{Rdata}} = 2$ and $\overline{num}_{\text{Vdata}} = 5$. Each set of data consists of 40 instances (La01-La40) and is divided into 8 different sizes. Brandimarte dataset consists of 10 instances (MK01-MK10), collectively representing 7 different sizes. These instances of different sizes feature varying numbers of jobs and machines ($n \times m$).

During DOAGNN training, we generate instances of each size for training and validation, retaining the policy that performs best on the validation set. This policy is then tested on the corresponding sizes in Hurink and Brandimarte datasets to assess DOAGNN's performance on out-of-distribution instances. In the generated instances, processing times range from [1, 20], the number of compatible machines per operation ranges from [1, $m$], and the number of operations per job ranges from [4, 12]. The training data is randomly sampled from these intervals.

## 5.2 Experiment Results and Analysis

In this subsection, we analyze DOAGNN's performance, evaluate its generalization capability, and conduct ablation studies on both the model and reward function.

*5.2.1 Performance comparison.* We present the results of the experiments in Table 1 and Table 2, where "LB" denotes the optimal or best-known solution for each instance. The full performance of DOAGNN on the Hurink dataset is detailed in Appendix D. The tables show the makespan for each instance, with values in parentheses representing the relative difference from the LB. The best results for each dataset are highlighted in bold.

Firstly, as shown in Table 1, RL-based methods generally outperform traditional dispatching rules due to their ability to learn more sophisticated scheduling policies. DOAGNN's results stand out, averaging a 7.16% improvement in solution quality over existing RL-based methods. This significant margin highlights the competitiveness and robustness of the scheduling policies learned by DOAGNN, further reinforcing its effectiveness in solving FJSPs. Although HGNN [37] and MPGN [21] models achieve better performance on certain datasets, they are at a disadvantage on datasets with more complex data compositions (e.g., Hurink dataset Vdata). This is because a single GNN struggles to balance the capture of features representing each state and the consideration of long-term benefits, whereas the dual-model architecture of DOAGNN utilizes two GNNs to learn these aspects separately.

As shown in Table 2, DOAGNN surpasses the four baselines across three datasets with varying operation compatibilities. To facilitate a better comparison of the strategies learned by the models, the results of DOAGNN and all RL-based baselines are obtained using a greedy strategy. The Vdata dataset, which presents a greater challenge due to a higher average number of machine choices per operation, further emphasizes the complexity of the scheduling problem. DOAGNN's marked superiority on this dataset highlights its capability in handling more complex scenarios. This effectiveness is largely attributed to the mechanism of MR-GNN, which embeds comprehensive machine information into operation features, thereby enhancing the model's ability to accurately capture

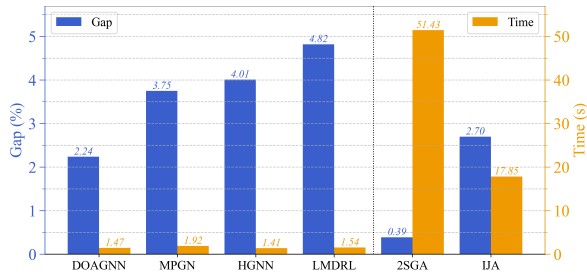

**Figure 4: Performance and runtime of our DOAGNN on the Hurink dataset Vdata.**

**Table 2: Performance of our DOAGNN on the Hurink dataset.**

|  | Edata | Rdata | Vdata |
|---|---|---|---|
| LB | *1028.9* | *934.3* | *919.5* |
| MWKR×EET | 1492.8(45.09%) | 1172.7(25.52%) | 992.6(7.95%) |
| MWKR×LWT | 1482(44.04%) | 1174.7(25.73%) | 1028.7(11.88%) |
| HGNN [37] | 1201.4(16.77%) | 1032.9(10.55%) | 956.4(4.01%) |
| LMDRL [51] | 1187.9(15.45%) | 1041.8(11.51%) | 963.8(4.82%) |
| **DOAGNN** | **1169.2(13.64%)** | **1015.6(8.7%)** | **940.1(2.24%)** |

and leverage the relationships between operations and machines during decision-making.

*5.2.2 Run Time Analysis.* As shown in Figure 4, we presents the performance and the average time of DOAGNN compared to three RL-based methods and two heuristic methods on the Hurink dataset Vdata. It is evident that heuristic methods explore the solution space to find higher-quality solutions; however, they consume more time compared to RL-based methods. DOAGNN retains the advantage of rapid inference associated and achieves excellent results within a reasonable time frame.

*5.2.3 Generalization Ability.* We conduct a generalization experiment using Hurink dataset Vdata la31-35 instances of size $30 \times 10$. To validate the model's ability to generalize to large-sized and unknown instances, we train DOAGNN on instances of size $10 \times 5$, then test the resulting policy on Vdata la31-35. Both HGNN [37] and LMDRL [51] are trained on instances of size $30 \times 10$, and the resulting policies are tested on Vdata la31-35. As shown in Table 3, our DOAGNN outperforms all baselines in the generalized results. Notably, we adjust the parameter shapes based on the number of machines during the basis decomposition (see Eq. (9)), which necessitates the use of zero-padding for the parameter $W_r$. By effectively adapting most parameters, knowledge from previous instances is successfully transferred to unknown instances, demonstrating the superior generalization capability of our DOAGNN.

*5.2.4 Ablation studies.* We conduct ablation studies on Brandimarte dataset to evaluate the effectiveness of our DOAGNN. In these experiments, we separately retained either the OS-GNN or the MR-GNN components of DOAGNN and also tested using raw state information without embedding it through our GNNs. As shown in Table 4, the solutions obtained using raw state information are inferior to other models, indicating the crucial role of incorporating machine information and the transmission of operation data. These

**Table 3: Generalization performance of our DOAGNN.**

| Vdata | la31 | la32 | la33 | la34 | la35 |
|---|---|---|---|---|---|
| LB | *1520* | *1658* | *1497* | *1537* | *1549* |
| MWKR×EET | 1591(4.7%) | 1737(4.8%) | 1588(6.1%) | 1582(2.9%) | 1600(3.3%) |
| MWKR×LWT | 1618(6.5%) | 1781(7.4%) | 1574(5.1%) | 1645(7%) | 1681(8.5%) |
| MPGN [21] | 1561(0.7%) | 1693(2.1%) | 1531(2.3%) | 1562(1.6%) | 1574(1.6%) |
| HGNN [37] | 1565(3%) | 1714(3.4%) | 1529(2.1%) | 1574(2.4%) | 1612(4.1%) |
| LMDRL [51] | 1575(3.6%) | 1726(4.1%) | 1531(2.3%) | 1594(3.7%) | 1575(1.7%) |
| **DOAGNN**$^*_{30 \times 10}$ | 1552(2.1%) | **1668(0.6%)** | **1511(0.94%)** | **1555(1.2%)** | **1564(1%)** |
| **DOAGNN**$_{10 \times 5}$ | **1549(1.9%)** | 1691(2%) | 1519(1.5%) | **1555(1.2%)** | 1573(1.6%) |

\* : DOAGNN$_{30 \times 10}$ is trained on instances of size $_{30 \times 10}$, while DOAGNN$_{10 \times 5}$ is trained on instances of size $_{10 \times 5}$.

**Table 4: Ablation studies of DOAGNN on the Brandimarte dataset.**

| BASELINE | LB | MR-GNN | OS-GNN | RawFeatures | **DOAGNN** |
|---|---|---|---|---|---|
| AVERAGE | *172.7* | 187.4(8.5%) | 189.7(9.8%) | 191.2(10.7%) | **185.6(7.4%)** |

**Table 5: Ablation studies of our reward function on Brandimarte dataset.**

| BASELINE | LB | $r_{tgt}$ | $r_{ave}$ | $r_{util}$ | $r_{t|a|u}$ |
|---|---|---|---|---|---|
| AVERAGE | *172.7* | 194.7(12.7%) | 209.1(21.1%) | 205.6(19.1%) | **185.6(7.4%)** |

findings also highlight the effectiveness of the dual-network architecture in DOAGNN for enhancing training quality. Additionally, MR-GNN outperformed OS-GNN in scheduling performance, likely because, although information transmission within operations is limited, the operation features alone can only partially compensate for the absence of machine information, which is critical for making informed machine selection decisions.

Additionally, we conduct ablation studies to evaluate the performance of using different reward functions (i.e. $r_{tgt}$, $r_{ave}$, $r_{util}$ and $r_{tgt} + r_{ave} + r_{util}$ (abbreviated as $r_{t|a|u}$)) on the Brandimarte dataset. Notably, $r_{tgt}$ is a commonly used reward function in the scheduling problem, following [37, 51, 53]. As the results presented in Table 4, our reward $r_{t|a|u}$ outperforms the other rewards, highlighting the effectiveness of incorporating the objectives of maximizing completion time and reducing machine utilization into our reward function. We present the results and analysis of the differing sparseness of reward $r_{tgt}$ and $r_{t|a|u}$ in Appendix E.

## 6 Conclusion

We propose a model named DOAGNN with a new reward to solve FJSPs. The dual-model architecture of DOAGNN can better balance the capture of features representing each state and the improvement of long-term benefits. Furthermore, our reward function reduces estimation errors and alleviates the issue of sparse rewards. Experimental results demonstrate that our DOAGNN exhibits excellent performance and shows remarkable generalization capabilities.

Although the offline training mode is often adopted in practical applications of FJSP, the dual-model architecture poses a relatively high computational cost during training. In future work, we consider directly predicting scheduling trends based on relationships between state changes, thereby reducing the training cost.

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

## A Details of state

The operation and machine states are encoded by the information from their respective nodes. In this paper, these state encodings are used as initial features input to the DOAGNN. We detail the composition of the operation and machine states as follows.

The meaning of each dimension in the state encoding for each operation is as follows: 1 if the operation has been scheduled, otherwise 0; the number of available machines for scheduling; the number of unscheduled operations within the same job; the estimated/actual start time (see Eq. (1)); the average processing time on all available machines; the estimated/actual end time (see Eq. (2)).

The meaning of each dimension in the state encoding for each machine is as follows: the end time of the nearby scheduled operation (the current total processing time of the machine); the number of schedulable operations; the ratio of machine idle time to its current total processing time; and the ratio of current total processing time to the maximum processing time of all machines.

## B Reward analysis

We analyze the components to be added to the reward based on the differences in two scheduling results. We present two scheduling results on the la05 dataset of Hurink Dataset Vdata [18] in Figure 5: one generates by our model (labeled as A) and the other obtains through dispatching rules MWKR×EET (labeled as B). As

shown, the results demonstrate that the scheduling result of group A significantly outperforms that of B. In scheduling result B, there exists a significant gap in completion time across different machines, with machines ($M_2$, $M_3$, $M_4$, $M_5$) remaining partially idle during the scheduling. Based on these findings, under the holistic scheduling goal of $\min(C_{max})$, we consider simultaneously reducing machine idle time and increasing both maximum and minimum completion time. Reducing machine idle time encourages the agent to increase machine utilization when selecting machines. However, relying solely on this objective causes the agent to overlook the varying processing time across different machines. This results in the agent ignoring machines with shorter processing time but lower utilization rates, leading to repetitive processing on a subset of machines and neglecting the overall scheduling scenario. Consequently, we propose a objective to increase the minimum completion time, which promotes diversity in machine selection and ensures a more even distribution of processing operations across all machines. However, focusing only on increasing the minimum completion time results in neglecting the overall scheduling scenario and potentially underutilizing certain machines. Therefore, both the minimization of maximum completion time and the maximization of machine utilization should be considered as complementary objectives.

## C The setting of parameters

We present the hyperparameter settings used in our experiment are shown in Table 6.

**Table 6: Basic settings of our experiments.**

| | |
|---|---|
| batch size | 64 |
| dimension of hidden layers in actor/critic networks | 64 |
| dimension of hidden layers in GNNs | 128 |
| the ratio of rewards:$\lambda_1 : \lambda_2 : \lambda_3$ | 1:0.8:0.3 |
| optimizer | Adam |
| initial learning rate | 0.0003 |
| the number of blocks in matrix factorization:B | 3 |
| clipping ratio in $clip$:$\epsilon$ | 0.2 |
| decay parameter in GAE:$\lambda$ | 0.9 |
| node dimensions embedded in GNNs | 12 |
| operations/machines node initial feature dimension | 10/5 |
| maximum number of iterations | 300 |
| discount factor in discounted return:$\gamma$ | 0.95 |
| discount factor in OS-GNN:$\gamma_o$ | 0.9 |

In a complete MDP, the total value obtained by $r_{tgt}$ corresponds to the maximum makespan, while the total value obtained by $r_{ave}$ corresponds to the minimum completion time. For a gradually converging model, the $r_{tgt}$ values fluctuate slightly within the neighborhood of $\frac{\min(C_{ij})}{\sum n_i}$, whereas the $r_{ave}$ values fluctuate within the neighborhood of $\frac{\max(C_{ij})}{\sum n_i}$. The gap between $r_{tgt}$ and $r_{ave}$ varies depending on the dataset and the quality of the scheduling results, generally not exceeding 10%. The sensitivity of $r_{util}$ exhibits differences across various datasets. In the Hurink dataset Vdata, complex selections enable machines to have stronger compatibility

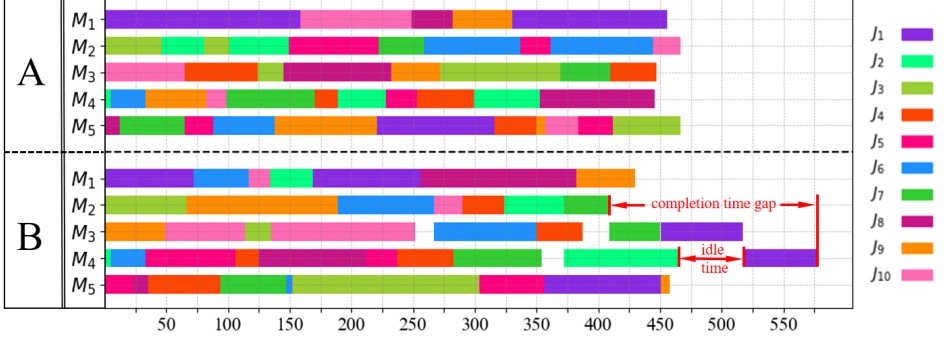

**Figure 5: Gantt charts of the two scheduling results on the Hurink Dataset Vdata la05.**

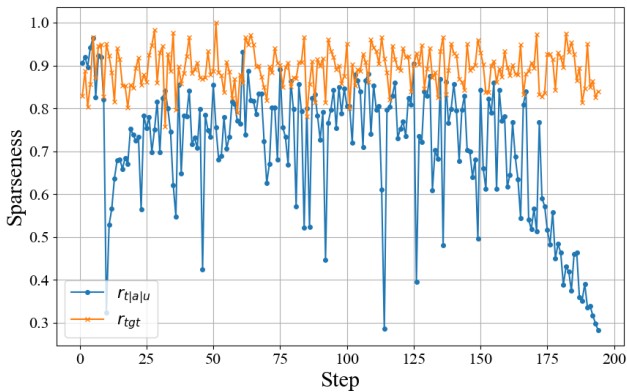

**Figure 6: Sparseness of different reward functions.**

$L_2$ norm of $R$ [17], as follows:

$$\text{Sparseness}(R) = \frac{\sqrt{\bar{n}} - (\sum_{x=1}^{\bar{n}} |R_x|)/\sqrt{\sum_{x=1}^{\bar{n}} R_x^2}}{\sqrt{\bar{n}} - 1}, \quad (20)$$

where $\bar{n}$ is the dimensionality of the reward vector $R$, and $R_x$ represents the $x$-th element in $R$. Sparseness($R$) ranges from $[0, 1]$, where a higher value indicates greater sparsity of $R$. Specifically, we define sparseness($R$) to be 1 when $R$ is a zero vector.

Figure 6 shows that our reward function reduces sparseness to approximately 1/3 of $r_{tgt}$ as the steps progress, indicating more consistent and informative feedback during the learning process. This advantage results from the inclusion of $r_{ave}$ and $r_{util}$, which align the reward more closely with scheduling objectives, leading to better-guided decision-making by the agent.

with operations, resulting in a machine idle rate approaching 0 in most high-quality scheduling results. Consequently, increasing the weight of $r_{util}$ ($\lambda_3$) can better guide the scheduling process. In contrast, in the Brandimarte dataset MK10, the idle rate disparities among scheduling results of varying quality remain minimal, necessitating a decrease in $\lambda_3$. To ensure better compatibility across all datasets, we set the ratios of $\lambda_1 : \lambda_2 : \lambda_3$ to 1:0.8:0.3.

## D  Performance of DOAGNN on the Hurink dataset

The complete performance of DOAGNN on the Hurink dataset is shown in Table 7. Due to the space limit, we represent the models HGNN [37], LMDRL [51] and MPGN [21] as $RL_1$, $RL_2$ and $RL_3$, respectively. The dispatching rules MWKR×EET and MWKR×LWT are referred to as $rule_1$ and $rule_2$, respectively.

## E  Analysis of reward sparsity

Additionally, we evaluate the performance of the reward $r_{tgt}$ and $r_{t|a|u}$ in terms of sparsity. Our experiment is conducted on 100 FJSP instances, each with a size of 20 × 10 and a total of 194 operations (i.e., 194 steps). The sparsity of the reward is computed based on the reward vector $R$ obtained by the agent at each step. We define sparsity by measuring the difference between the $L_1$ norm and the

## Table 7: Performance of our DOAGNN on the Hurink dataset.

| $n \times m$ | | | | Edata | | | | | | Rdata | | | | | | | Vdata | | | |
|---|---|---|---|---|---|---|---|---|---|---|---|---|---|---|---|---|---|---|---|---|
| | | LB | ours | $RL_1$ | $RL_2$ | $rule_1$ | $rule_2$ | LB | ours | $RL_1$ | $RL_2$ | $rule_1$ | $rule_2$ | LB | ours | $RL_1$ | $RL_2$ | $RL_3$ | $rule_1$ | $rule_2$ |
| | la01 | 609 | **621** | 688 | 658 | 866 | 882 | 571 | 617 | **616** | 663 | 709 | 687 | 570 | **599** | 609 | 612 | 610 | 660 | 642 |
| | la02 | 655 | **771** | 851 | 809 | 982 | 992 | 530 | **555** | 598 | 666 | 694 | 695 | 529 | **544** | 582 | 601 | 555 | 662 | 587 |
| $10 \times 5$ | la03 | 550 | **653** | 657 | 660 | 692 | 988 | 478 | 520 | **507** | 525 | 623 | 565 | 477 | **512** | 517 | 533 | 532 | 544 | 595 |
| | la04 | 568 | 674 | **660** | 669 | 872 | 836 | 502 | 563 | **533** | 573 | 633 | 624 | 502 | 544 | 554 | 556 | **530** | 598 | 659 |
| | la05 | 503 | 541 | **530** | 593 | 737 | 599 | 457 | **476** | 504 | 500 | 540 | 507 | 457 | **491** | 493 | 525 | 507 | 578 | 614 |
| Average Gap | | | **12.99%** | 17.36% | 17.47% | 43.81% | 48.94% | | **7.6%** | 8.66% | 15.32% | 26.04% | 21.27% | | **6.11%** | 8.67% | 11.51% | 7.85% | 20% | 22.17% |
| | la06 | 833 | **895** | 926 | 951 | 1030 | 955 | 799 | **836** | 850 | 856 | 912 | 848 | 799 | **817** | 859 | 888 | 820 | 890 | 938 |
| | la07 | 762 | **890** | 910 | 896 | 1065 | 1063 | 750 | **776** | 787 | 825 | 846 | 908 | 749 | 768 | 785 | 802 | **757** | 816 | 811 |
| $15 \times 5$ | la08 | 845 | **900** | 969 | 947 | 1267 | 1162 | 765 | **801** | 804 | 820 | 960 | 829 | 765 | **773** | 805 | 783 | 782 | 848 | 853 |
| | la09 | 878 | **912** | 1046 | 946 | 1229 | 1085 | 853 | **882** | 875 | 885 | 1068 | 940 | 853 | 874 | **862** | 887 | 879 | 931 | 939 |
| | la10 | 866 | 888 | **883** | 886 | 1167 | 1032 | 804 | 854 | **832** | 870 | 973 | 920 | 804 | **824** | 839 | 843 | 862 | 867 | 871 |
| Average Gap | | | **7.19%** | 13.14% | 10.56% | 37.62% | 26.6% | | 4.48% | **4.45%** | 7.17% | 19.84% | 11.93% | | **2.16%** | 4.53% | 5.86% | 3.27% | 9.62% | 11.13% |
| | la11 | 1103 | 1211 | **1172** | 1213 | 1451 | 1274 | 1071 | **1083** | 1135 | 1131 | 1157 | 1156 | 1071 | **1079** | 1114 | 1119 | 1101 | 1153 | 1117 |
| | la12 | 960 | 1041 | **1001** | 1065 | 1282 | 1154 | 936 | **952** | 957 | 977 | 1050 | 1086 | 936 | 968 | 967 | 977 | **950** | 1015 | 1014 |
| $20 \times 5$ | la13 | 1053 | **1150** | 1215 | 1164 | 1392 | 1270 | 1038 | 1060 | 1107 | **1051** | 1120 | 1127 | 1038 | **1051** | 1076 | 1082 | 1053 | 1084 | 1136 |
| | la14 | 1123 | **1163** | 1191 | 1234 | 1447 | 1487 | 1070 | **1106** | 1107 | 1110 | 1161 | 1201 | 1070 | 1103 | 1099 | 1088 | **1086** | 1113 | 1119 |
| | la15 | 1111 | **1297** | 1301 | 1297 | 1541 | 1389 | 1090 | **1133** | 1228 | 1236 | 1202 | 1282 | 1089 | **1098** | 1104 | 1110 | 1111 | 1111 | 1152 |
| Average Gap | | | **9.57%** | 9.9% | 11.64% | 32.95% | 22.87% | | **2.47%** | 6.32% | 5.76% | 9.31% | 12.43% | | **1.82%** | 2.99% | 3.3% | 1.86% | 5.22% | 6.41% |
| | la16 | 892 | **1010** | 1059 | 1051 | 1351 | 1414 | 717 | **815** | 794 | 821 | 984 | 968 | 717 | 731 | 753 | 727 | 717 | 767 | 726 |
| | la17 | 707 | 819 | 781 | **773** | 1039 | 1016 | 646 | 750 | 731 | **720** | 794 | 825 | 646 | 674 | 658 | 648 | **647** | 659 | 728 |
| $10 \times 10$ | la18 | 842 | 939 | 944 | **934** | 1089 | 1153 | 666 | 784 | 814 | **766** | 920 | 888 | 663 | 663 | 663 | 663 | 663 | 680 | 704 |
| | la19 | 796 | 981 | 950 | **972** | 1100 | 1234 | 700 | 839 | 884 | **812** | 950 | 1023 | 700 | 650 | 654 | 639 | **626** | 647 | 757 |
| | la20 | 857 | **974** | 1007 | 1004 | 1317 | 1199 | 756 | 883 | 943 | **878** | 1117 | 1001 | 756 | 766 | **756** | 766 | **756** | 766 | 807 |
| Average Gap | | | **15.36%** | 15.8% | 15.63% | 44.01% | 46.94% | | 16.81% | 19.54% | **14.69%** | 34.86% | 35% | | 2.5% | 2.5% | 1.29% | **0.294%** | 3.53% | 9.5% |
| | la21 | 1017 | 1264 | **1243** | 1256 | 1363 | 1594 | 835 | **946** | 989 | 967 | 1096 | 1146 | 804 | 840 | **839** | 891 | 887 | 926 | 1043 |
| | la22 | 882 | 1038 | **977** | 1070 | 1310 | 1208 | 760 | 876 | **857** | 893 | 1072 | 1132 | 736 | **768** | 812 | 809 | 793 | 817 | 857 |
| $15 \times 10$ | la23 | 950 | **1072** | 1131 | 1137 | 1396 | 1394 | 842 | **947** | 956 | 1010 | 1137 | 1146 | 815 | **837** | 868 | 872 | 858 | 895 | 957 |
| | la24 | 909 | **1047** | 1112 | 1086 | 1527 | 1360 | 808 | 937 | 958 | **928** | 1095 | 1085 | 775 | **794** | 803 | 873 | 883 | 879 | 904 |
| | la25 | 941 | **1059** | 1139 | 1102 | 1401 | 1629 | 791 | **888** | 895 | 939 | 1172 | 1130 | 756 | **807** | 809 | 840 | 883 | 877 | 926 |
| Average Gap | | | **16.62%** | 19.21% | 20.26% | 48.9% | 52.9% | | **13.82%** | 15.33% | 17.36% | 38.05% | 39.71% | | **4.11%** | 6.3% | 10.26% | 10.75% | 13.07% | 20.61% |
| | la26 | 1125 | **1290** | 1338 | 1348 | 1647 | 1674 | 1061 | **1155** | 1173 | 1175 | 1292 | 1406 | 1054 | **1064** | 1073 | 1110 | 1089 | 1145 | 1165 |
| | la27 | 1186 | **1390** | 1485 | 1423 | 1863 | 1916 | 1091 | **1182** | 1225 | 1214 | 1314 | 1482 | 1084 | **1115** | 1133 | 1131 | 1123 | 1171 | 1233 |
| $20 \times 10$ | la28 | 1149 | 1348 | 1441 | **1344** | 1769 | 1609 | 1080 | **1159** | 1199 | 1175 | 1343 | 1422 | 1070 | **1079** | 1100 | 1106 | 1106 | 1171 | 1191 |
| | la29 | 1118 | **1331** | 1353 | 1359 | 1741 | 1807 | 998 | **1070** | 1101 | 1111 | 1387 | 1273 | 994 | **1019** | 1027 | 1076 | 1049 | 1178 | 1138 |
| | la30 | 1204 | 1423 | 1459 | **1410** | 1846 | 1863 | 1078 | **1200** | 1227 | 1275 | 1361 | 1382 | 1069 | **1087** | 1106 | 1143 | 1117 | 1223 | 1215 |
| Average Gap | | | **17.29%** | 22.38% | 19.05% | 53.33% | 53.39% | | **8.62%** | 11.62% | 12.09% | 26.16% | 31.21% | | **1.76%** | 3.18% | 5.59% | 4.04% | 11.32% | 12.73% |
| | la31 | 1539 | 1751 | **1720** | 1755 | 2170 | 2153 | 1521 | **1598** | 1636 | | 1829 | 1840 | 1520 | **1552** | 1565 | 1575 | 1561 | 1591 | 1618 |
| | la32 | 1698 | 1934 | 1962 | **1904** | 2530 | 2490 | 1659 | **1732** | 1798 | 1765 | 1913 | 1931 | 1658 | **1668** | 1714 | 1726 | 1693 | 1737 | 1781 |
| $30 \times 10$ | la33 | 1547 | **1706** | 1776 | 1785 | 2396 | 2228 | 1499 | 1557 | 1568 | **1542** | 1891 | 1700 | 1497 | **1511** | 1529 | 1531 | 1531 | 1588 | 1574 |
| | la34 | 1604 | 1898 | 1821 | **1817** | 2183 | 2349 | 1536 | 1596 | **1577** | 1686 | 1747 | 1767 | 1537 | **1555** | 1574 | 1594 | 1562 | 1582 | 1645 |
| | la35 | 1736 | **1911** | 1967 | 2080 | 2547 | 2498 | 1550 | **1616** | 1692 | 1706 | 1773 | 1764 | 1549 | **1564** | 1612 | 1575 | 1574 | 1600 | 1681 |
| Average Gap | | | **13.24%** | 13.81% | 14.98% | 45.56% | 44.23% | | **4.3%** | 5.93% | 7.34% | 17.87% | 15.93% | | **1.14%** | 3% | 3.09% | 2.06% | 4.34% | 6.93% |
| | la36 | 1162 | 1367 | 1334 | **1295** | 1675 | 1775 | 1030 | 1247 | 1296 | **1229** | 1372 | 1411 | 948 | 958 | 993 | 973 | 985 | **976** | 1106 |
| | la37 | 1397 | 1582 | 1650 | **1531** | 1987 | 2177 | 1077 | 1243 | **1210** | 1294 | 1488 | 1528 | 986 | 1010 | 1069 | 1024 | 1028 | **1006** | 1155 |
| $15 \times 15$ | la38 | 1144 | 1379 | 1507 | **1370** | 1810 | 1690 | 962 | **1098** | 1128 | 1107 | 1389 | 1516 | 943 | 955 | 945 | **943** | 948 | 981 | 1069 |
| | la39 | 1184 | **1357** | 1547 | 1412 | 1757 | 1884 | 1024 | **1153** | 1174 | 1228 | 1452 | 1455 | 922 | **936** | 951 | 942 | 979 | 1026 | 1054 |
| | la40 | 1150 | **1293** | 1353 | 1311 | 1877 | 1798 | 970 | 1139 | 1098 | **1108** | 1435 | 1392 | 955 | 956 | 985 | 967 | 968 | **965** | 1066 |
| Average Gap | | | 15.58% | 22.42% | **14.61%** | 50.83% | 54.44% | | **16.13%** | 16.65% | 17.83% | 40.94% | 44.22% | | **1.28%** | 3.97% | 1.99% | 3.23% | 4.2% | 14.64% |

