# OpenReview forum: "Dual Operation Aggregation Graph Neural Networks for Solving Flexible Job-Shop Scheduling Problem with Reinforcement Learning"
_ACM.org/TheWebConf/2025/Conference — WWW 2025 Poster_

### Official Review · Reviewer_vibt · 2024-11-07

**Novelty:** 6
**Technical Quality:** 6

**Review:**

The DOAGNN model proposed by the authors for the Flexible Job-shop Scheduling Problem seems interesting and would add great value to the body of research in improving efficient decision-making in cloud manufacturing.

Quality
The quality of the paper is very good and the mathematical expressions of the DOAGNN from the Markov decision process to the decomposition and improvement of the disjunctive graphs are well explained and relevant to understand how it will improve decision-making in cloud manufacturing.

Clarity
The authors did clarify all equations and mathematical expressions for the DOAGNN model.

Originality
The authors did make a comprehensive on the related works highlighting limitations and their proposed solution. This shows that the model presented in the paper is original and can be accepted by any body of research.

Significance
Based on the performance results of the experiments, the authors showed that the DOAGNN model significantly improves efficient decision-making in terms of machines and operation in cloud manufacturing.

Strengths
1. The abstract of the paper is very clear and well-presented.
2. A comprehensive literature review of the flexible job-shop scheduling problem for manufacturing processes based on traditional and reinforcement learning methods.
3. The Contributions of the paper are well explained and performance results are also clearly presented.

Weakness
1. A summary of the major related works including their limitations and proposed solution by the DOAGNN is not presented.
2. The authors did not provide a very clear reason why their proposed model is far better than other sequencing models in terms of decision-making.

**Questions:**

1. During the review of the related works, did you come across other dynamic graph-based modeling techniques that can be used for the Machine-to-operation graph and operation-to-operation graph?

2. In terms of computational cost, do you think the DOAGNN can be applied to a real cloud manufacturing environment to improve the efficient decision-making process?

3. Based on the performance results, you showed that other RL-based models performed better than DOAGNN, please explain why your proposed model is still better and should be accepted to improve decision-making.

**Reviewer Confidence:**

4: The reviewer is certain that the evaluation is correct and very familiar with the relevant literature

**Scope:**

4: The work is relevant to the Web and to the track, and is of broad interest to the community

---

### Official Review · Reviewer_Cqmn · 2024-11-20

**Novelty:** 4
**Technical Quality:** 4

**Review:**

### Summary:

The paper proposes DOAGNN to solve the Flexible Job-Shop Scheduling Problem (FJSP) using reinforcement learning. By decomposing disjunctive graphs into M2O and O2O subgraphs, the model captures relationships between machines and operations more effectively. A tailored reward function addresses reward sparsity and estimation bias.

### Pros:

1. The decomposition into M2O and O2O graphs effectively reduces graph density, enabling clearer representation of relationships and improved scheduling efficiency.
2. The multi-objective reward function significantly enhances decision-making and addresses reward sparsity issues.
3. DOAGNN achieves superior performance and strong generalization across diverse benchmarks.

### Cons:

1. The dual-model structure demands significant computational resources during training, which may limit scalability and raise practical concerns about balancing performance gains against the increased costs. It should be necessary to make the trade-off analysis.
2. The reliance on offline training may limit the ability to adapt to some real-world dynamic environments. Please explain the necessity of the setting in the paper in detail.
3. The experiments of hyperparameter sensitivity should be included.

**Questions:**

Refer to Cons.

**Reviewer Confidence:**

2: The reviewer is willing to defend the evaluation, but it is likely that the reviewer did not understand parts of the paper

**Scope:**

4: The work is relevant to the Web and to the track, and is of broad interest to the community

---

### Official Review · Reviewer_3Tw7 · 2024-11-25

**Novelty:** 4
**Technical Quality:** 5

**Review:**

This paper proposed two improvements over existing RL-based methods for the problem of Job-Shop Scheduling Problem (FJSP) and it demonstrated an overall state of art performance on public benchmark datasets. Overall the paper presented the problem definition clearly (which is a well-defined problem by itself) and presented their modifications to the existing RL-based methods with clarity. Overall i think this is a valuable addition to solving the problem of FJSP.

The major drawbacks are:
1. The lacks a fundamental justification of decoupling the disjunctive graph into two separate graphs. In the writing it talks a lot more about the "how" then the "why". I think this part can be discussed a bit further.
2. Lack of more comprehensive ablation studies: the authors only performed ablation on one dataset; however, there are a lot of design choices involved in this paper, such as the separation of the disjunction graph and the choice of one GNN encoder and a cross-modality GNN module; if the empirical results were shown for the different design choices across all datasets, this paper would make a much stronger case for the specific choices it makes.
3. The experiment section argues that the DOAGNN model is better than HGNN and MPGN since it is able to work better on the "more complex" Hurink dataset; however, more backgrounds need to be provided in their benchmark section for people who have not worked on this problem before.
4. possible typo in L285, where E should perhaps be D.

Overall this is a solid work but not an excellent one, and I think it is a valuable contribution to the subcommunity working on this problem.

**Questions:**

- I would very much like to see the ablation on both datasets and how the performance changes.
- I would very much like to see the writing improvements for people who are not working on this particular problem.

**Reviewer Confidence:**

3: The reviewer is confident but not certain that the evaluation is correct

**Scope:**

3: The work is somewhat relevant to the Web and to the track, and is of narrow interest to a sub-community

---

### Official Review · Reviewer_bn5C · 2024-12-02

**Novelty:** 4
**Technical Quality:** 4

**Review:**

This paper studied the Flexible Job-Shop Scheduling Problem (FJSP). The authors propose a method called DOAGNN, which is based on reinforcement learning and Graph Neural Networks (GNNs) to address this issue. They decouple the disjunctive graph into two distinct graphs and develop two distinct graph aggregation methods, along with a designed reward function.

**Pros:**
1. The explanation of the Markov Decision Process is clear, which makes the problem statement precise.
2. Experiments were conducted on datasets of varying scales and compared against multiple types of baselines.
3. The model design is appropriately integrated with the problem, and the design of the two GNNs (MR-GNN and OS-GNN) takes into account the specific task scenario.

**Cons:**
1. The introduction of the GNN component is overly complex and difficult to understand. The authors should focus more on explaining the methods of information aggregation rather than the common GNN structures.
2. The paper lacks ablation studies for two critical components of the model: graph construction and GNN network design. It remains unclear what effects would be observed if the simplest GNN, such as a Graph Convolutional Network (GCN), were employed following the graph construction method in the paper.

**Questions:**

1. You mentioned that ‘reinforcement learning-based methods fail to learn relationships between FJSP nodes’. However, in Section 2.2, you refer to existing RL methods that integrate GNNs [4, 15, 27, 37, 47, 54]. Does this viewpoint lack rigor?
2. In the design of the reward function, how are the hyperparameters $\lambda_i$ that control the contributions of the three reward components determined?
3. In real-world task scenarios, what does a time difference of 1 second versus 20 seconds signify? Is such an improvement in speed worth the trade-off in performance?

**Reviewer Confidence:**

3: The reviewer is confident but not certain that the evaluation is correct

**Scope:**

4: The work is relevant to the Web and to the track, and is of broad interest to the community

---

### Official Review · Reviewer_7LA3 · 2024-12-03

**Novelty:** 4
**Technical Quality:** 4

**Review:**

In this paper, the authors are concerned with the flexible
Job-shop Scheduling Problem (FJSP), in particular in the context
of cloud manufacturing. In an effort to propose an efficient
and adaptive solution for FJSP, the authors introduce a dual
operation aggregation graph neural network (DOAGNN), which is
based on a disjunctive graph, as these graphs can indicate the
relationships between operations and machines within a JFSP
instance; here, the disjunctive graph is split into two
distinct graphs to improve certain aspects of efficiency and
effectiveness. Some other improvements take place, and the
method is shown to outperform the state of the art.

This work goes well beyond my expertise, but at least I can
say that I find the idea of using one graph for capturing
the relationships among machines and operations (viz., M2O
graph), and another graph for the relationships among different
operations (viz., O2O graph) rather intuitive. Due to the use of
various existing techniques, I am not too convinced about the
novelty, but I will monitor the discussion closely.

**Questions:**

In the conclusion there is a statement about the high computational
cost during offline training, but where do we see that in the paper?

**Reviewer Confidence:**

1: The reviewer's evaluation is an educated guess

**Scope:**

3: The work is somewhat relevant to the Web and to the track, and is of narrow interest to a sub-community